# Interaction of Neural Stem Cells (NSCs) and Mesenchymal Stem Cells (MSCs) as a Promising Approach in Brain Study and Nerve Regeneration

**DOI:** 10.3390/cells11091464

**Published:** 2022-04-26

**Authors:** Agnieszka Kaminska, Klaudia Radoszkiewicz, Paulina Rybkowska, Aleksandra Wedzinska, Anna Sarnowska

**Affiliations:** Translational Platform for Regenerative Medicine, Mossakowski Medical Research Institute, Polish Academy of Sciences, Pawinskiego 5, 02-106 Warsaw, Poland; akaminska@imdik.pan.pl (A.K.); kradoszkiewicz@imdik.pan.pl (K.R.); prybkowska@imdik.pan.pl (P.R.); awedzinska@imdik.pan.pl (A.W.)

**Keywords:** neural stem cells, mesenchymal stem cells, niche, coculture, cell interaction, nervous system regeneration

## Abstract

Rapid developments in stem cell research in recent years have provided a solid foundation for their use in medicine. Over the last few years, hundreds of clinical trials have been initiated in a wide panel of indications. Disorders and injuries of the nervous system still remain a challenge for the regenerative medicine. Neural stem cells (NSCs) are the optimal cells for the central nervous system restoration as they can differentiate into mature cells and, most importantly, functional neurons and glial cells. However, their application is limited by multiple factors such as difficult access to source material, limited cells number, problematic, long and expensive cultivation in vitro, and ethical considerations. On the other hand, according to the available clinical databases, most of the registered clinical trials involving cell therapies were carried out with the use of mesenchymal stem/stromal/signalling cells (MSCs) obtained from afterbirth or adult human somatic tissues. MSCs are the multipotent cells which can also differentiate into neuron-like and glia-like cells under proper conditions in vitro; however, their main therapeutic effect is more associated with secretory and supportive properties. MSCs, as a natural component of cell niche, affect the environment through immunomodulation as well as through the secretion of the trophic factors. In this review, we discuss various therapeutic strategies and activated mechanisms related to bilateral MSC–NSC interactions, differentiation of MSCs towards the neural cells (subpopulation of crest-derived cells) under the environmental conditions, bioscaffolds, or co-culture with NSCs by recreating the conditions of the neural cell niche.

## 1. Introduction

Currently, there is no effective therapy to repair and restore the function of the central nervous system (CNS). The discovery of newly generated cells in the adult brain was a breakthrough in neurobiology as it provided irrefutable arguments that the processes of adult neurogenesis in the brain are actually happening. The term ‘neurogenesis’ means literally ‘the birth of the neurons’ and includes neurogenesis at prenatal (embryonic stage) and postnatal (adulthood) age. Adult neurogenesis is the process of the formation, migration, maturation, and integration of new neurons in the brain of adult mammals, including humans. So far, two active neurogenic zones in the CNS have been distinguished: the subventricular zone (SVZ) of the lateral ventricle, located along the ependymal cell layer, and the subgranular zone (SGZ) of the dentate gyrus of the hippocampus. These areas are perfect examples of neural stem cells (NSCs) displaying heterogeneity: SVZ generates olfactory bulb interneurons and corpus callosum oligodendrocytes while SGZ develops dentate granule neurons and astrocytes [1]. However, researchers discovered more areas in the adult brain where new neurons may occur, including the hypothalamus, the striatum, the amygdala, periventricular zones, and the frontal and temporal cortex [2,3].

Providing the niche—a comprehensive environment containing growth, nutritional, and regulatory factors—is necessary to form new neurons. Injury of the nervous system can activate stem cells residing in the niche, leading to the young neuroblast migration to the site of damage [4,5]. Researchers attempt to translate this phenomenon to the clinical therapy. The idea of endogenous neurogenesis activation collapsed because of a limited number of newly generated cells which proved to be insufficient to achieve a clinical effect. In the next attempts, the allogeneic NSCs were transplanted near the site of damage to increase neuroblasts; however, the same problems still arose, i.e., poor cell survival and low proliferation. It seems that the failure of NSC transplantation is related to the lack of cell niche factors that protect and support newly generated cells. Mimicking the multifactorial niche would therefore improve the NSCs therapeutic efficiency.

To create a functional microenvironment an additional, a supportive type of cells is required. Mesenchymal stem/stromal/signalling cells (MSCs) appear to play a supportive role through their paracrine and immunomodulatory properties (Figure 1). MSCs can migrate to damaged sites of the brain where they produce neuroprotective and vasoprotective effects [6,7]. Furthermore, they can promote cell regeneration in injured cerebral cortex [8], while their secretome has a favourable impact on neurons after traumatic brain injury [9]. However, research is still underway as to whether MSCs may interact with NSCs or their niche to increase the regenerative capacity of NSC differentiation towards mature and functional neurons. MSCs and NSCs alone have a low regenerative potential in the CNS, but their unique features can create a synergistic relationship that is essential for therapeutic effect enhancement. While NSCs could provide new neural progenitors, MSCs could immunoprivilege the microenvironment and support the survival and differentiation of newly generated neuroblasts through the adjuvant effect and cell-cell contact [10] (Figure 1). It was recently reported that both cell types could have beneficial effects; however, the mechanism responsible for their therapeutic properties seems to be different from what was observed in the animal model of Alzheimer’s disease [11]. The implementation of such a therapy requires a lot of further research and understanding of the mechanisms controlling the processes which take place in the natural niche. In this review, we summarise the knowledge of both neural and supportive cells and their interactions.

### Neural Stem Cells Niche

The complex environment comprising many types of cells and the extracellular matrix (ECM), as well as the signalling molecules associated with each population of stem cells, is collectively termed as the stem cell niche. The niche is a local microenvironment that maintains self-renewal capacity and the potential for multidirectional NSC differentiation [12].

A stem cell niche provides a protective environment that regulates proliferation, differentiation, and apoptosis to control stem cell reserves [13,14]. Therefore, maintaining a balance between the quiescence and activity of stem cells is a feature of a functional niche [15]. Different signals and components mediate the communication between different cell types that constitute the niche. The NSC niche also contains other types of cells, such as radial glia, neuroblasts, neural progenitor cells, neurons, astrocytes, and ependymocytes as supportive cells [16]. Moreover, vasculature also plays an important role through endothelial cells and pericytes with their paracrine factors [17,18]. In the NSC microenvironment, the major role of ECM focuses on neural development and regeneration processes, such as neurogenesis, nerve repair, neuronal cell migration, and axonal growth. The biophysical properties of ECM that are of importance for its function include elasticity, matrix pore size, structure, and topography. The interactions between the aforementioned niche components are depicted in Figure 2.

The behaviour of the neural cell niche in pathological conditions is not fully understood, which is why ongoing research is trying to improve our understanding of these mechanisms in order to design therapies which shall not only support the regeneration of the damaged brain but also the entire central nervous system.

The differentiation toward a more specialised form of NSCs could be controlled by the modulation of the niche-constructing factors. Extracellular ligands are a key component of the regulatory niche necessary to maintain NSCs in the process of self-renewal, differentiation, cell adhesion, and migration. Some of many extracellular factors associated with cell specification and cell signalling include basal fibroblast growth factor (FGF), epidermal growth factor (EGF), CXCL12, stromal cell-derived factor 1 (SDF-1), Sonic hedgehog (Shh), bone morphogenic proteins (BMP), and Notch [19]. The regulation of these factors can help in the regeneration of damaged brain sites by selectively directing cells in a neuronal direction. Current research mostly focuses on the possibilities of increasing the differentiating potential towards mature, functional neurons. One of the methods of modification is to change ECM so as to limit the differentiation in the direction of glia or astrocytes in favour of the differentiation into neurons. In recent reports, Wang et al. constructed a neuronal matrix in which they anchored the neurotrophic factors: brain-derived neurotrophic factor (BDNF), nerve growth factor (NGF), and chondroitinase ABC [20]. The produced matrix implanted in post-brain-damage mice reduced the glial furrow and increased the neurogenic zone in the brain. The results of that study proved how important it was to supply the cell niche with the factors determining the future of NSCs, i.e., both those supporting neurogenesis and the ones eliminating NSC differentiation into microglia cells.

One of the problems is the invasiveness of surgical methods that affect the CNS. The possibility of providing favourable factors by other means than intracerebral administration of a therapeutic agent capable of migrating to the niche of neural stem cells seems to be a potential solution. To obtain signal enrichment from the NSC niche, the unique properties of mesenchymal stem cells can be used.

## 2. Neural Stem Cells

NSCs in the adult brain are present in two states: they can be active or quiescent, which relates to two important niche-controlling processes, neurogenesis, and quiescence, respectively.

Quiescent NSCs (qNSCs) are characterised by a low metabolic rate and high sensitivity to environmental signals. Unlike stem cells derived from other tissues, NSCs undergo multiple stages of activation at any time [21,22], which maintains the balance between proliferation and quiescence, limits the metabolic stress, and, as a result, restrains senescence and transformation processes. It regulates the rate of neurogenesis and neurogenic capacity of the human brain [23]. In order to prevent NSCs from premature exhaustion, qNSCs present the expression of the cyclin-dependent kinases—p57, p27, and p21. A similar role was shown for chromodomain helicase DNA-binding protein 7(CHD7) [24,25,26,27]. NSC quiescence maintenance is also provided by a balanced activation of canonical and non-canonical WNT activity. In SVZ, it is induced by a non-canonical signalling pathway via Rho-GTPase CDc42 activation, promoting anchorage to the niche and Notch1, as well as the N-cadherin upregulating expression [28]. Recently, several studies have investigated the role of circadian rhythms on the proliferation and differentiation of adult neural stem cells. A circadian clock regulates metabolic processes; thus, it can limit the exhaustion of qNSCs by restraining their activation into the cell cycle or, conversely, wake them up [29,30].

The activation of qNSCs is described as the first stage of neurogenesis. Neurogenesis and gliogenesis are strictly controlled by a plethora of synergistic and antagonistic factors, such as cytokines, morphogens, or neurotransmitters of the niche microenvironment (Figure 3). Morphogens are critical for tissue specification during embryonic and adult CNS development. Fundamental examples of morphogens include: BMP signalling, which regulates the neurogenesis by promoting astroglial commitment of NSCs as well as their quiescence [31]; Notch signalling, which induces the maintenance and proliferation of NSCs, also involved in brain damage and repair processes of stroke, Alzheimer’s disease, and CNS tumours [32,33]; Wnt, which promotes quiescence and self-renewal of NSCs [34]; and Shh, which is responsible for NSC maintenance [35,36]. 

NSCs secrete several soluble factors, such as growth and neurotrophic factors or cytokines, which assert the protection of existing neural cells and the replacement of the lost ones. bFGF and RA increase the potential to differentiate into neurons and PDGF strengthens oligodendrocyte development, while EGF, CNTF, and bone morphogenic proteins (BMPs) enhance differentiation into astrocytes [13,37,38] (Figure 3). Although neural precursor cells (NPCs) give rise to all three cell types in CNS, they are not multipotential during early stages of gestation—they differentiate only toward neurons. Then, they can differentiate into astrocytes and oligodendrocytes. Those processes are strictly regulated externally by cytokines, as well as via internal cell programs, such as epigenetic modification [39,40,41,42,43,44]. The members of interleukin-6 (IL-6) family, including leukaemia inhibitory factor (LIF) or ciliary neurotrophic factor (CNTF), as well as retinoic acid, have been shown to jointly induce the astrocytes generation from NPCs [39]. Other growth factors involved in adult neurogenesis regulation also help to promote the proliferation and neuron generation of NPCs. Additionally, IGF was reported to inhibit BMP signalling, thus stimulating differentiation of NPCs into oligodendrocytes [45,46]. 

Adult neurogenesis is strongly regulated by microglia, as well as by inflammatory processes. In pathological conditions, the reparatory mechanism of neurogenesis in the adult brain cannot be activated as it is inhibited by microglia that releases, such cytokines as IL-6 and tumour necrosis factor—α (TNF-α). It has been shown that microglia can promote neurogenesis when anti-inflammatory molecules prevail in the niche, or when physical exercise is performed [47,48,49,50]. It has been reported that, surprisingly, the cellular accumulation of reactive oxygen species can actually promote cell proliferation and survival [51,52,53]. Consistently, several available reports have already shown that NSC treatment with exogenous reactive oxygen species sufficiently promoted the proliferation of self-renewal of NSCs due to the activation of PI3K/AKT pathway [54].

Furthermore, NSCs have been shown to possess a significant ability to migrate, even across long distances, and integrate to the injured regions of the brain of all ages [55]. The migration is possible thanks to the SDF-1 up-regulation in astrocytes and endothelial cells of injured tissue and its CXC receptor 4 (CXCR_4_) on NSC surfaces [56].

### Current Challenges in Neural Stem Cells Therapy

It has been demonstrated that endogenous NSCs can respond to brain injuries and help in its repair. NSCs can administrate the niche by paracrine and autocrine signals. Although it is hard to determine the niche signals origin, several studies have shown the impact of the factors released by NSCs on the niche. For example, upon an increased level of intracellular calcium, NSCs release vasolidating factors that activate pericytes to increase blood flow [57]. NSCs can promote qNSC activation by expressing vascular endothelial growth factor receptor-3 (VEGFR-3) and its ligand VEGF-C [58].

In recent years, a growing body of evidence has pointed to the immense importance of NSCs in terms of their application in cell therapy. Currently, as many as 74 clinical studies are registered on clinicaltrials.gov. There is a steadily growing interest in the application of these cells in order to treat neurological diseases, including ischemic stroke, Parkinson’s and Alzheimer’s diseases, amyotrophic lateral sclerosis (ALS), or spinal cord injury (SCI). As endogenous the restorative abilities of NSCs are ineffective, a plethora of sources on NSCs have been tested, including foetal and adult CNS-derived NSCs, neural progenitor cells from pluripotent stem cells, as well as non-neural stem cells (Table 1). So far, no ideal source available has been selected [59]. Each of these sources has its pros and cons, which is summarised in the table below.

## 3. Mesenchymal Stem Cells (MSCs)

MSCs are a highly heterogeneous group of cells characterised by rapid proliferation capacity, phenotypic plasticity, and multi-differentiation into functional cell types. Along with embryonic stem cells (ESCs) and induced pluripotent stem cells (iPSs), which are also extensively studied, they are currently one of the most prosperous and promising therapeutic methods in regenerative cell-based therapies.

The most significant asset for which MSCs are frequently used in today’s medicine is their relatively high availability in the niches of the adult organism. The major sources of mesenchymal cells include: bone marrow (BM-MSCs) [62], adipose tissue (AD-MSCs) [63], placenta [64], umbilical cord tissue, mainly Wharthon’s jelly (WJ-MSCs) [65], umbilical cord blood (UCB-MSCs) [66], dental pulp [67], muscles [68], dermis [69], and many others. The acquisition of MSCs is considered as a readily available, uncomplicated, and safe source of cells. Moreover, there are no ethical dilemmas which accompany the acquisition of ESCs or NSCs. The therapeutic benefits of MSCs can also be obtained by combining cells with other structures, such as biomaterials or scaffolds, which seems to be an innovative method in injuries repair [70].

### 3.1. Supportive Mechanism of MSCs

MSCs can significantly influence the regeneration of damaged tissues due to the ability to repopulate at the site of injury, immunomodulatory capacity, and paracrine activity, which consists of secretory activity and exosome release. Through intercellular contacts, MSCs can receive appropriate stimuli signals to independently create identical copies of themselves through mitotic division, and then to migrate, repair, and replace damaged cells [71,72]. This communication is enabled by key mediators which are known as exosomes or extracellular vesicles (EVs) [73]. Those naturally released liposomes, mainly filled with proteins, lipids, nucleic acids, and even organelles, are involved in the intercellular transportation of their contents to maintain physiological homeostasis [74]. Current research shows that EVs isolated from MSCs can be an alternative to standard transplantation technology of MSCs because of numerous advantages, such as greater safety or lower immunogenicity. Some reports also show that EVs may support regeneration in animal models of stroke [75], brain [76], and spinal cord injuries [77], but it is unlikely that they will have greater therapeutical effectiveness than the standard MSC therapy [78]. What is important, however, is that exosomes can overcome the blood–brain barrier and interact with nerve cells. In order to enhance properties of exosomes, their combinations with various nanoparticles are used. Some observations of exosomes in a mouse model of Alzheimer’s disease showed that they migrated towards inflammation sites, but got dispersed when entering the bloodstream in non-inflammation conditions. Furthermore, it was observed that exosomes enriched with gold nanoparticles were selectively captured by neurons [79]. The combination of MSC-derived exosomes and magnetic nanoparticles increased the desirability of exosomes for ischemic stroke. In addition, they exhibited an increased secretion of therapeutic agents, thanks to which they promoted anti-inflammatory response, angiogenesis, and antiapoptosis [80].

An ability to interact with the immune system components is another feature that enables MSCs to effectively regenerate tissue after damage [81]. MSCs suppress the immune response by changing the cytokine profile of T-cells to anti-inflammatory ones [82]. MSCs also interact with monocytes and macrophages [83,84], modulate B-cell functions [85], or inhibit T-cell proliferation [86]. Furthermore, some immunomodulatory factors secreted by MSCs, such as IL-6 or TNF-α, may function in both directions: anti-inflammatory or pro-inflammatory, depending on the external injury microenvironment [87]. MSCs can secrete cytokines such as IL-6, hepatocyte growth factor, prostaglandin 2, inducible nitric oxide synthase, and TGF-β1 that regulate CD4+ T-cell subsets, suppress immune response, and protect neurons from apoptosis [88,89].

Despite the fact that both processes—MSC proliferation and trafficking of exosomes between cells—play an extremely important role, what seems to be of key significance is the support of the injury repair provided by MSC secretome activity and a vast number of bio-active molecules, such as chemokines, growth factors, cytokines, and proteins, which are released to extracellular space [90]. These specifically released biomolecules perform a wide range of actions, activating or inhibiting some signalling pathways, stimulating or attenuating receptors, and affecting a range of properties. They can act as proangiogenic (VEGF, IL-6, and MCP-1) [91], anti-apoptotic, immunomodulatory (IL-6, IL-10, CCL-2) [92,93], or neuroprotective (NGF, BDNF, NT-3, NT-4/5, GDNF, LIF, and EGF) [94] factors, and may also be responsible for the migration of MSCs as well as other types of cells. Moreover, only the implantation or even just a presence of MSCs at the site of tissue damage may trigger the influx of other proteins, cytokines, or interleukins already present in the organism and associated with the stimulation of endogenous tissue repair processes.

### 3.2. Strategies to Increase the Neurogenic Potential of MSCs

In regenerative medicine, MSCs can be used as a source of neural cells or as a sup-port for NSCs. The neural transdifferentiation of MSCs can be achieved by providing signalling molecules. Neural phenotypes were acquired by MSCs after adding all-trans RA or 2-(3-ethylureido)-6-methylpyridine (UDP-4). Furthermore, UDP-4 was reported to increase specific neural genes and protein expression in MSCs [95]. Another strategy exploits the high heterogeneity of a MSC population containing NSC-like cells or cells which can differentiate into NSC-like cells. Some clones were found to exhibit neural phenotypse undifferentiated or/and exposed to the neural differentiation condition [96]. The same authors also reported that MSC population could be contaminated by NSCs [93]. In the presence of specific signals, MSCs lose typical markers and resemble NSCs, thereby acquiring neural morphology with expression of neural genes and proteins. Such differentiated MSCs were shown to be more effective in the treatment of experimental autoimmune encephalitis mouse model than nondifferentiated MSCs [97]. It was also reported that MSCs changed their secretory properties in the presence of neurogenic differentiation media and those upregulated cytokines are known to be essential for astrocytes development [98]. Except for the addition of components influencing neural differentiation to culture media, other strategies are also being looked for to increase neurogenic potential of MSCs. Here, we focused on two promising approaches—neurosphere culture and 3D culture with cellular scaffolds.

#### 3.2.1. Neurospheres from MSCs

Although neurosphere assay is one of the characteristic properties that help to identify such cells as NSCs, it does not mean that it is only unique for this stem cell population. Under favourable conditions, MSCs can form aggregates resembling neurospheres (Table 2). Many protocols for the sphere culture of MSCs use reagents similar to the neurosphere culture of NSCs, such as EGF, bFGF, and N2 or/and B27 supplements. Not all of the aforementioned components are necessary to establish neurospheres; however, the complete medium is the most efficient [99]. The neurosphere could be generated by MSCs derived from different sources. The umbilical cord appears to be the most efficient source in comparison to bone marrow and adipose tissue. Primary neurospheres derived from UC-MSCs were found to be larger than those from BM-MSCs and AD-MSCs, and they also formed more secondary spheres [100]. MSCs from different sources can form neurospheres in the same formula medium [99,100,101]. Time needed for neursphere culture formation may vary depending on the research groups; aggregates are formed even during the first 24 h. Viability of MSCs drastically decreases after the fourth day of culture, which is why it usually lasts no longer than 3 days in vitro. Long-term neurosphere cultures (3 weeks) of MSCs were reported to impair cell proliferation and clonogenicity while increasing senescent processes [102].

Neurosphere formation was found to increase the potency of the neural differentiation of MSCs into neuronal and glial linage [100,109]. The principle of inducing MSC neurosphere neural differentiation is similar to the differentiation of the NSC neurosphere. The sphere is seeded on a coated surface in medium with serum [100,103] or without mitogens [110]. MSCs cultured as spheres exhibit a higher expression of the neural progenitor, the mature neuron, and glial markers under neural differentiation than MSCs from standard culture [109]. However, some other papers reported a higher expression of neuronal markers, but no expression of glial marker GFAP in cells derived from the neurosphere upon neural differentiation [110]. With regards to their morphological and functional properties, the differentiated cells resembled neurons, astrocytes, and oligodendrocytes. Furthermore, neuron-like cells derived from AD-MSCs neurosphere displayed voltage-dependent sodium current with fast activation and fast inactivation [104].

However, some differences between MSC neurospheres and NSC neurospheres were observed. The transcriptional profile of UC-MSC neurospheres was found to include transcriptional profiles of both MSCs and NSCs [99]. A lot of research findings show that culture system of MSC neurospheres increases the expression of pluripotent markers, such as Nanog, Oct3/4, and Sox2 [99,101,109]. However, some other research groups did not confirm this observation [102,110]. A Japanese research team developed MSC aggregates called MUSE cells (multilineage-differentiating stress-enduring cells), which do not only express pluripotency markers but they also differentiate into cells of all three layers [113,114].

#### 3.2.2. 3D Neural Scaffold of MSCs

Scaffold generation is the next strategy to consider in order to increase the neural capacities of MSCs. The scaffold structure resembles the ECM environment as it mimics the cellular niche [115], which could enhance the potential differentiation of MSCs into neuroglial cells in vivo. The scaffold helps to graft cells directly to the injured area where cells could regenerate injured structures [116,117].

The scaffold properties can influence the differential potential of MSCs, and a wide range of approaches have already been tested (Table 3). Multiple research groups developed structures from various materials, including natural resources, such as chitosan, collagen, silk, or synthetic polymers (i.e., poly-L-lactide or graphene). The next initiative of scaffold functionalisation was to increase neural differentiation by molecules, such as growth factors [117,118] or short peptides [119]. Combining two compounds might also change the outcome, as the incorporation of carbon nanotubes was found to improve the neural potential of collagen scaffold for MSCs [120]. The alignment of fibres is another important property—linear polymers may orient the extension of regenerating axons [117,121]. Electric stimulation could improve therapeutic effects, which was observed as an increased expression of neural markers [122,123].

Neural scaffolds for MSCs appeared to be very promising in vivo. In spite of different approaches, transplantations of MSC scaffolds supported the regeneration of neural structures in rats and dogs with spinal cord injuries [116,126,128]. Neuron-like cells derived from MSC scaffolds were reported to integrate into the spinal cord and reconstruct neural circuits [116,128]. MSCs transplanted on collagen scaffolds were observed to direct polarisation of endogenous macrophages into anti-inflammatory M2 phenotypes [125]. Grafts did not only increase the number and survival rate of neural cells, but they also improved behavioural outcomes of injured animals [116,117,125,126].

### 3.3. Current Clinical Use of MSCs and Future Perspectives

There are many ambiguities in the use of MSCs in the treatment of neurological diseases. Some studies have shown several beneficial effects, while the opposite results have also been reported. MSCs have been extensively studied with regards to their application in the treatment of ischemic stroke. It has been found that, due to their plasticity, MSCs can differentiate into a neuron-like phenotypes and have enormous immunomodulatory potential [131]. The preclinical results seemed to be promising [132], but they also showed many side effects of the treatment, including embolism, creation of tumors, or even β-amyloid accumulation [133]. In addition, the safety and effectiveness of MSC administration routes into the injured nervous cells should be carefully considered and optimised in future clinical stem cell therapies [134,135]. Some studies have shown a huge potential of 3D spheroids in therapeutic applications as a novel strategy for cell therapy in strokes [136].

One of the most frequently discussed problems surrounding the clinical usefulness of MSC treatment is the time-limited capacity for therapeutically effective proliferation [137], as the potential for the intensive division of MSCs is known to decrease significantly above the seventh passage. After this transition, some morphological, phenotypic, and chromosomal changes appear, and the ageing process begins. MSCs can no longer regenerate and differentiate as intensively as they did in the early passages. These properties are also associated with the medium and conditions of cell cultures [138]. Another common, but equally important, factor regards the sensitivity of MSCs to environmental changes in oxygen concentration. Interestingly, 5% oxygen concentration hypoxia has been shown to significantly increase cell proliferation [139]. Other factors that have a huge impact on the therapeutic effectiveness of MSCs and affect their heterogeneous plasticity are related to the source of stem cells, age and inter-individual variability of the donors, and the procedure of tissue collection and isolation [140,141]. According to the current literature, phenotypic reprogramming of MSCs, depending on environmental changes during in vitro culture, can significantly modulate the regenerative properties as well as the safety of cell transplantation [142,143].

## 4. MSC and NSC Interaction

Interactions between differentiated MSCs and NSCs are observed in coculture systems which combine the benefits from both cell types (Table 4). NSC differentiation in vitro is improved by interaction with MSCs. Moreover, NSCs could also promote the neural differentiation of MSCs [144]. MSCs do not only influence the differentiation process, but also increase the proliferation and survival of NSCs [145,146]. Coculture with MSCs was shown to preserve the stemness of NSCs, even in the absence of EGF and bFGF in culture medium [145,147]. Cell-to-cell direct contact offers greater benefits than transwell culture or NSC culture in MSC-conditioned media [10]. In mixed neurosphere systems, the shell of the structure is formed by NSCs, whereas MSCs create the core [148]. Moreover, coculture can change the differentiation profile of MSCs [145] and the transcription profile of both stem cell types is modified during coculture as well [149].

MSCs can affect the differentiation of neuroglial cells as well. The coculture of MSC neurospheres and primary astrocytes was first found to induce synapse formation, and the observed structures (dendrites, cell bodies, and spines) were identified as parts of the neuron. Then, their electrical activity and action potential were detected [106]. Furthermore, MSCs were reported to rescue neural cells from apoptosis in a oxygen–glucose deprivation model [75,156]. Astrocytes in the ALS mice model were found to increase glutamate uptake after coculture with MSCs [154], and MSCs were shown to prevent a reduction in synaptic density in the in vitro Alzheimer’s disease model [155].

The observed therapeutic effects are associated with the secretion of trophic factors, both by MSCs and NSCs. MSCs were proven to modulate stressed neuronal cell survival by increasing the expression of anti-inflammatory cytokines (TGF-β, IL-6, and IL-10) and decreasing the expression of pro-inflammatory factors (NF-κB and COX-2) [156]. It was not only TGF-1 expression that was increased by MSCs, but also the expression of TGF-1 receptor in NSCs [149]. Moreover, an inhibition of TGFβ signalling blocked differentiation processes [149]. It was shown that thrombospondin-1 secreted by UCB-MSCs protected hippocampal neurons from synaptic density loss in the in vitro Alzheimer’s disease model [155]. In the presence of hippocampal culture slices, Wharton jelly and WJ-MSCs were found to enhance neuroprotective effects by secretion of neurotrophic factors (TGFβ and VEGF) and an increased expression of neurotrophic factors genes (GDNF and bFGF), as was the case with ADSC [75,157]. NSCs can also influence the neuronal differentiation of MSCs via BDNF and NGF secretion [144].

Numerous investigations showed that Notch-1 signaling was involved in the interaction between MSCs and NSCs [145,148,149]. However, some researchers reported that Notch-1 did not regulate differentiation-induced expression of neuroglial markers in ADSC [158]. Others suggested that the Wnt–MSC signaling pathway increased the neurogenesis of NPCs in the Alzheimer’s disease model by stimulating the Wnt pathway [152]. MSCs reduced the concentration of intracellular calcium [Ca2+] and generation of reactive oxygen species in stressed neurons, which resulted in decreased neuronal apoptosis [156].

The co-transplantation of MSCs with NSCs exerted a therapeutic effect in vivo. Due to their immunomodulatory activity, MSCs provided the environment for grafted NSCs [159], which enhanced NSC survival in vivo [146,148,160]. The presence of MSCs reduced the number of NSCs required for graft [148], resulting in a greater functional recovery in rats after spinal cord injury [160], and prolonging therapeutic benefits in rats with Huntington disease [159]. The transplantation of SDF1-overexpressing NSCs and MSCs was reported to improve behavioural functions. [147]. However, co-transplantation of MSCs and NSCs resulted in tumour formation, whereas tumours were not observed in NSC- and MSC-only groups [147].

## 5. Conclusions

The interaction between stem cells and companion/supporting cells is crucial for homeostasis and tissue regeneration. In many organs, MSCs sustain the survival of stem cells and the proliferation of transit-amplifying cells (TACs), which then differentiate into target cell types. Although there is still insufficient evidence, it seems that mesenchyme as a component of the cell niche may play a role in the homeostasis between the population of quiescent SCs and the population of progenitors that proliferate, differentiate, and migrate from cell niche. The same interactions are described between NSCs and MSCs. As NSCs are potentially an unlimited source of all neural cell types and MSCs display a high paracrine activity, their combined therapeutic use for neurological disorders seems to be highly prospective. MSCs can be considered as a source of NSC-like cells as they can form neurospheres. Nevertheless, most available studies do not provide enough strong evidence, as neural potential is measured through early neural markers, such as nestin, β-III-Tubulin, or GFAP. The possibility of MSC differentiation into mature neuronal or glial cells is still a topic of discussion. However, their function in therapy seems to be different given the supportive and regulative components of NSC niches. Both NSCs and MSCs have distinct genetic programs that complement each other and form an inseparable unit to maintain tissue homeostasis.

Based on the available scientific literature, the combination of MSCs and NSCs appears to represent a promising therapeutic prospect (Figure 4). NSCs could be delivered as a source of cells that differentiate into mature neural and glial cells. MSCs, in turn, could support the transplantation of NSCs through neuroprotective, immunomodulatory, and pro-angiogenic activity. An additional advantage in the use of MSCs is the ease of isolation from multiple sources and the lack of tumourigenesis. Although the mechanisms of this mutual interaction remain unexplored, the effects are already very promising. Nonetheless, more research into the use of this combination is still needed, especially in animal models and human clinical trials (Figure 4).

## Figures and Tables

**Figure 1 cells-11-01464-f001:**
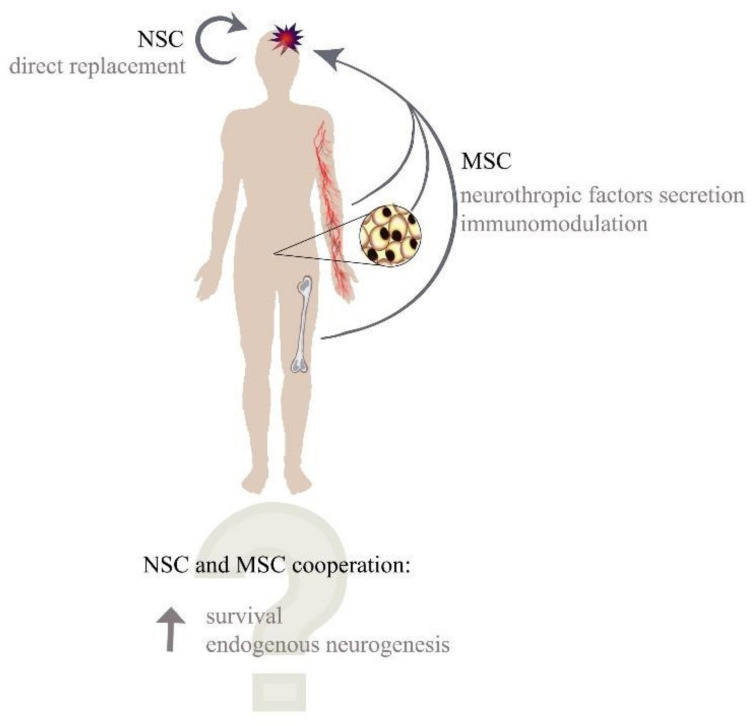
Potential abilities of MSC and NSC cooperation.

**Figure 2 cells-11-01464-f002:**
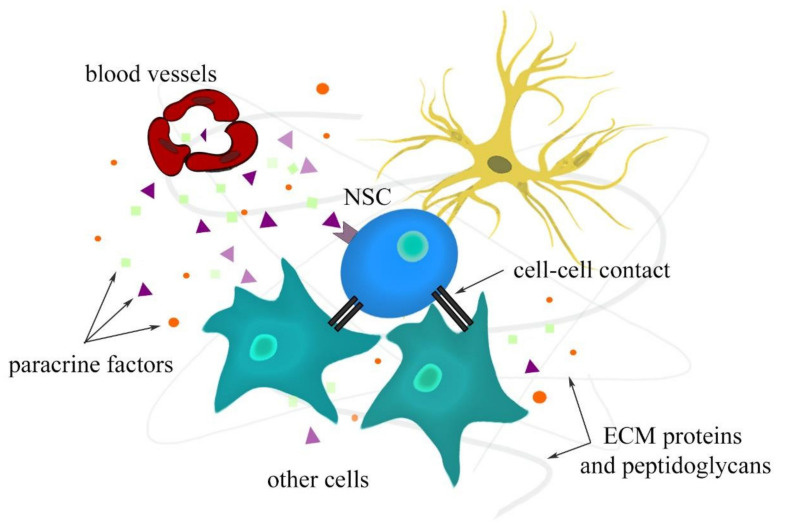
Interactions in neural stem cell niche.

**Figure 3 cells-11-01464-f003:**
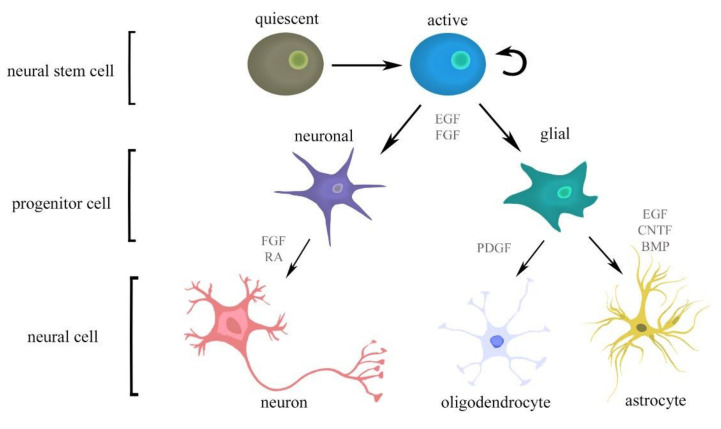
Activation and differentiation of NSC processes are controlled by multiple signalling molecules. Abbreviations: bone morphogenic proteins (BMPs), ciliary neurotrophic factor (CNTF), epidermal growth factor (EGF), fibroblast growth factor (FGF), platelet-derived growth factor (PDGF), and retinoic acid (RA).

**Figure 4 cells-11-01464-f004:**
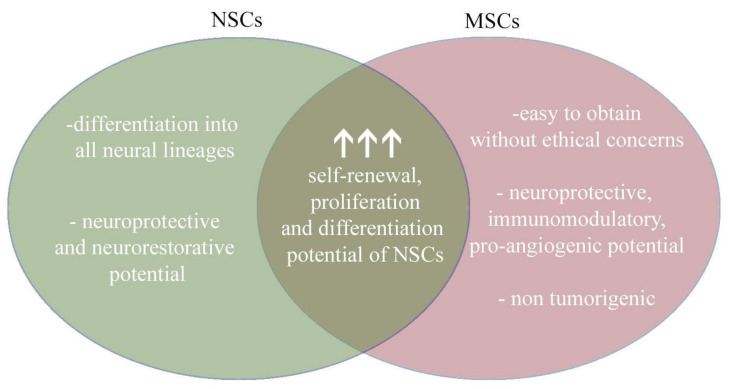
Combining the therapeutic benefits of NSCs and MSCs provide a new perspective for therapy.

**Table 1 cells-11-01464-t001:** The summary of potential advantages and disadvantages according to use of different neural stem cells sources in cell-replacement therapy [14,55,60,61].

Source of NSCs	Advantages	Limitations
Fetal CNS	primary, undifferentiated cells extensive proliferation potential limited differentiation potential	availability limited numer of cells safety-considerable genetic instability
Adult CNS	differentiation potential into functional neural cells genetic stability compatibility to microenvironment of the brain	availability safety limited proliferation potential
ESCs	unlimited proliferation potential pluripotency	ethical issues safety concerns—high tumorigenic risk limited availability risk of unwanted transdifferentiation
iPSCs	availability unlimited proliferation potential pluripotency can be expanded indefitely in vitro (in theory)	safety concerns—possible genetic instability risk of unwanted transdifferentiation
Other sources	availability from many sources safety extensive proliferation potential extensive paracrine activity (neurotrophic effect) autologous tranplantations genetic stability	limited proliferation and differentiation potential—poor direct neuroregenerative effect

CNS—central nervous system, ESCs—embryonic stem cells, iPSCs—induced pluripotent stem cells, NSCs—neural stem cells.

**Table 2 cells-11-01464-t002:** Methods of forming MSC neurospheres.

MSCs Source	EGF	FGF	Supplement	Additional Conditions	Bibliography
adipose tissue bone marrow Wharton jelly	+	+	N2	*Ns*	[100]
adipose tissue	+	+	-	*Ns*	[103]
adipose tissue	1. − 2. − 3. +	1. + 2. − 3. +	1. − 2. + 3. +	Three different media (medium 1 and 2 contained L-glutamin and β-mercaptoehanol)	[104]
adipose tissue	+	+	B27	*Ns*	[105]
adipose tissue	+	+	N2, B27	*Ns*	[101]
bone marrow	+	+	-	Low attachment surface	[106]
bone marrow	+	+	N2, B27	*Ns*	[96]
breast milk	+	+	N2, B27	*Ns*	[107]
dental pulp	+	+	N2, B27	*Ns*	[108]
umbilical cord	+	+	-	Low attachment surface	[109]
umbilical cord	+	+	N2, B27	*Ns*	[99]
Wharton jelly	+	+	B27	Presence of heparin in medium	[110]
Wharton jelly	1. + 2. +	1. − 2. +	1. N2 2. B27	Low attachment surface, cultured for 21 days	[102]
Wharton jelly	1. + 2. +	1. − 2. +	1. N2 2. B27	Nonadherent conditions	[111]
Wharton jelly	1. + 2. +	1. − 2. +	1. N2 2. B27	*Ns*	[112]

Abbreviations: EGF—epidermal growth factor, FGF—fibroblast growth factor; 1. First medium, than changed to 2. Medium. *Ns*—further conditions were not specified.

**Table 3 cells-11-01464-t003:** Neural scaffolds for MSCs.

Scaffold Material	MSCs Source	Additional Information	Observed Result	Bibliography
carbon nanotubes	bone marrow (c)	Single/multi-COOH group addition	- Increased expression of neuronal markers in vitro	[124]
collagen hydrogel + carbon nanotubes	bone marrow (r)	*Ns*	- Increased secretion of neurotrophic factors compared to 2D conditions - Increased expression of neural markers compared to collagen scaffold	[120]
chitosan	umbilical cord (h)	BDNF incorporation	- Any toxic effect observed - BDNF released by scaffold for 30 div	[118]
collagen	bone marrow (r)	*Ns*	- Increased survival rate in vivo and improved behavioural outcomes - Activation of M2 anti-inflammatory macrophages in vivo	[125]
collagen	placenta (h)	Linear ordered fibres	- Promotion of axonal regeneration, synapse formation and remyelination in vivo	[121]
collagen	umbilical cord (h)	*Ns*	- Promotion of endogenous neurogenesis, neuron maturation, remyelination, and synapse formation in vivo - Improved locomotor recovery	[117,126]
fibrin or platelet lysate	Wharton jelly (h)	Hydrogels consisted of fibrin or platelet lysat with 5% or 21% of oxygen in the atmosphere	- Increased expression of neural markers compared to 2D conditions - Increased expression of neurotrophic factors - Reduced mortality of hippocampal cells under oxygen–glucose deprivation	[127]
gelatine sponge	bone marrow (c)	Genetically modified MSCs overexpressing TrkC—receptor for NT3, coculture with Schwann cells overexpressing NT3	- Differentiation into neuron-like cells with electrophysiological function and formation of synapse structures in vitro - Regeneration of nerve tract in vivo, - Motor function improvement in paralysed limb	[116]
gelatine sponge	bone marrow (r)	Genetically modified MSCs overexpressing NT-3 and receptor of NT-3: TrkC	- Differentiation into neural-like cell in vitro - Transdifferentiation into myelin-forming cells in vivo - Promotion of host axonal regeneration and survival of host neurons in vivo	[128]
PLGA	bone marrow (r)	*Ns*	- Expression of MAP2 by MSCs in vitro and in vivo	[129]
PLGA nanofibers	dental pulp (h)	Aligned and nonaligned fibres, NGF incorporation	- Upregulation of nestin expression	[115]
PLLA nanofibers	conjunctiva (h)	Electrospinning	- Expression of neurocytes marker	[130]
rGO+PEDOT	rat (ns)	Provided electric stimulation by triboelectric nanogenerator	- Enhanced proliferation of MSCs - Improved neural differentiation	[123]
silk fibroin	bone marrow (h)	Integrin-binding laminin peptide motifs (YIGSR, GYIGSR) incorporation	- Promotion of MSCs stemness - Induction of neural differentiation in neural culture medium	[119]
silk fibroin, rGO	conjunctiva (h)	Electrical stimulation with 100 Hz or 0.1 Hz	- Increased expression of neuronal markers under 100 Hz stimulation	[122]

Abbreviations: c—canine, h—human, r—rat, ns—not specified; BDNF—brain-derived neurotrophic factor, NGF—nerve growth factor, NT3—neutrotrophin 3, PEDOT—poly3,4-ethylenedioxythiophene, PLGA—poly(lactic-coglycolic) acid, PLLA—poly-L-lactic acid, rGO—reduced graphene oxide, TrkC—tropomyosin receptor kinase C.

**Table 4 cells-11-01464-t004:** Effects of MSC coculture with neural cells in vitro.

Neural Cells Source	MSCs Source	Additional Information	Observed Result	Bibliography
**NSCs and NPCs**				
Adult brain (m)	AT (r)	Coculture as spheres, on chitosan surface	- Promotion of NSC survival in vitro and in vivo	[148]
Adult hippocamp, ventrical zones (r)	BM (r)	Adherent culture of BM-MSCs and NSCs	- Increased oligodendral differentiation of NSCs	[150]
Adult hippocamp (r)	BM (h)	NSCs cultured over MSCs	- Stimulation of NSC differentiation into astrocytes and oligodendrocytes	[149]
Brain (m)	AT (m)	NSCs were irradiated before, and the transwell system coculture was used with MSCs	- Higher survival of irradiated NSCs after coculture - Higher clonogenicity of irradiated NSCs after coculture	[151]
Hippocampal NPCs (r)	MSCs, ns (h)	*Ns*	- NPCs treated with amyloid-β (Alzheimer disease model) - Increased neurogenesis of treated NPCs and enhanced neuronal differentiation	[152]
Whole brain extracts (r)	BM (r)	Lack of mitogens in medium	- Preservation of NSCs stemness	[147]
Fetal tissue (h)	BM (h)	NSCs over MSCs or MSCs over NSCs, With or without Notch-1	- Increased expression of Notch-1 and Hes-1 by NSCs - Increased proliferation of NSCs - Enhanced stemness of NSCs	[145]
Fetal tissue (h)	BM (h)	Transwell system coculture	- Promotion of neuronal differentiation of BM-MSCs - Increased NGF and BDNF secretion	[144]
Cell line (ATCC, Catalog #CRL-2925), (m)	AT (h)	Mixed coculture, in vitro ischemia model	- Inhibition of NSCs apoptosis	[146]
NPC cell line (Millipore) (r)	WJ termed and pretermed (h)	Direct coculture and transwell coculture	- Increased expression of glial markers of NPCs in direct coculture	[10]
NSCs derived from iPSCs (h)	AT (h)	Culture inserts, MSCs over NSCs	- Prevention of lipopolysaccharide-induced activation of nuclear factor-κB (NF-κB) in NSCs - Smaller scars and better preservation of β-III tubulin-positive axons after transplantation of NSCs and MSCs to rats with spinal cord injuries	[153]
**Differentiated cells**				
Fetal brain astrocytes (h)	BM (h)	MSC neurosphere	- Synapse formation - Generation of electrically active neurons	[106]
Astrocytes (m)	AT (h)	Astrocytes derived from the ALS mice model, Transwell system coculture	- Enhanced glutamate uptake - Increased secretion of neuroprotective agents	[154]
Fetal hippocamp, neurons (m)	UCB (h)	Transwell system coculture, neurons treated with Aβ42 (the Alzheimer disease model)	- Prevention of reduction in synaptic density caused by Aβ42 peptide in the Alzheimer disease model	[155]
Neurons differentiated from SH-SY5Y (h)	UCB (h)	OGD-stressed neurons, MSCs inserts	- Rescue of neuronal cells from apoptosis	[156]
Organotypic hippocampal slices (r)	WJ (h)	OGD-stressed hippocampal slices	- Neuroprotective effect of MSCs - Enhanced neural differentiation of WJ and WJ-MSCs	[75]
Organotypic hippocampal slices (r)	WJ (h)	OGD-stressed hippocampal slices, transwell system coculture	- Decreased apoptosis and vascular atrophy of hippocamp	[6]

Abbreviations: h—human, m—mouse, r—rat; AT—adipose tissue, BDNF—brain-derived neurotrophic factor, BM—bone marrow, iPSCs—induced pluripotent stem cells, MSCs—mesenchymal stem cells, NGF—nerve growth factor, NPCs—neural progenitor cells, NSCs—neural stem cells, OGD—oxygen glucose deprivation, UCB—umbilical cord blood, WJ—Wharton jelly.

## Data Availability

Not applicable.

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
