# Peer review of "Interaction of Neural Stem Cells (NSCs) and Mesenchymal Stem Cells (MSCs) as a Promising Approach in Brain Study and Nerve Regeneration"

_cells, 2022, doi:10.3390/cells11091464_

Round 1

Reviewer 1 Report

Agnieszka KamiÅ„ska et al.  

It’s hard to know where to begin and end comments on this review as it covers so much ground in the very rapidly moving field of neural stem cell biology.  So I hope you will allow me to make some very general comments that might be helpful.  

The first point is that this manuscript has not been proof read.  There are two very basic aspects to this comment.  First, that there are a few clear typographical omissions.I will illustrate this by drawing your attention to section 3.2.2 where the heading needs to be adjusted to read 3-D neural scaffold of MSCs.  Perhaps one of your editorial team could look through it. They could check for this type of issue and for the second basic editorial issue, much of the text does not read in the normal idiom of English.  From. that same section you will find many sentences that don’t follow normal grammatical structures.  I believe this will distract interested readers from the value of this text.

The authors make a serious attempt to summarize the nature and utility of mesenchymal stem cells.  Although there are many places where I find the text verges one the hyperbolic, the reader can gain insight from following the authors logic.  In closing this brief comment, my view is this manuscript would benefit from both editorial and conceptual attention before publication.

Author Response

Thank you for your important remarks. We have corrected the topographical “mess” which was the result of working in two different computer systems (Libra and Office). We apologize for that.

Also, because of language problems, the paper was sent for professional linguistic correction.

We have made every attempt to introduce the suggested corrections (in blue) into the manuscript and we hope that you will approve our paper. 

Reviewer 2 Report

The paper is an interesting and extensive review of NSG and MSG interaction and how these two cells can bring a change and add dimension to the current therapies for nervous system affliction.

However, a few points need to be addressed and fixed, before it is considered for further action.

  • The study can benefit from better English writing and grammar editing. A few examples from the study are:

Line 57 -58: typo error

Line 111: missing ref date and grammar issues

Line 181: spacing gap

Line 355: grammar .. in the case of treat the nervous system, and neurodegenerative diseases.

  • Unify the format for writing “Table and Figure” in some places it is mentioned as Tab and others as Table.
  • The authors seem to neglect some important results that have been recently achieved in this specific field such as: doi: 10.3390/biom12020218. doi: 10.4103/1673-5374.314323.  doi: 10.3390/life12030392.  doi: 10.1007/s12015-021-10321-9.
  • The new research-based advances in the field that have demonstrated promising results should be discussed in the review
  • The paper will benefit from an illustration depicting the possible mechanistic approaches and pathways involved in NSC and MSC interaction.
  • The figures are not properly discussed in the paper.
  • Since this is a new approach and studies are ongoing to explore and establish the MSC-NSC based treatment procedure, the review will benefit from incorporating a method section regarding the same.

Author Response

Thank you for your insightful revision. We have corrected all the mentioned mistakes. Also, because of language problems, the paper was sent for professional linguistic correction. We have also supplement the manuscript with important, suggested papers. The figure description was expanded.

We have not incorporated a separate method subsection, since most of the groups used modified, different methods. We fully agree that this is the extremely important problem. We have just submitted a separate paper only about neural stem cell culture methods (isolation, culture, cryopreservation), but the topic seems to be too broad for that manuscript.

 We have made every attempt to introduce all the suggested corrections (in blue) into the manuscript and we hope that you will approve our paper.